# Study of Microstructure and Fatigue in Aluminum/Steel Butt Joints Made by CMT Fusion-Brazing Technology

**DOI:** 10.3390/ma15072367

**Published:** 2022-03-23

**Authors:** Yu Fang, Shanglei Yang, Yubao Huang, Xuan Meng

**Affiliations:** Department of Material Processing Engineering, School of Materials Engineering, Shanghai University of Engineering Science, Shanghai 201620, China; fangyunoemie@163.com (Y.F.); huangyb16@163.com (Y.H.); mengxuan1818@163.com (X.M.)

**Keywords:** cold metal transfer, aluminum/steel butt joint, microstructure, fatigue

## Abstract

Cold metal transfer (CMT) fusion brazing technology was used to weld 6061 aluminum alloy and Q235 galvanized steel with ER4043 welding wire. The microstructure, hardness, tensile performance, and fatigue performance of the welded joint were observed and analyzed. The results show that the tensile strength of the welded joint is 110.83 MPa and the fatigue strength limit is 170 MPa. In the fatigue process, the coupon first undergoes cyclic hardening and then cyclic softening and a ratchet effect occurs. The coupon was broken at the interface layer or weld zone where the fatigue strength limit is the lowest. The fatigue crack initiation is mainly caused by: (1) inclusions and second-phase particles; and (2) porosity and incomplete fusion. When cracks encounter holes during expansion, the expansion direction will change. The fatigued coupon displays a toughness fracture in the instantaneous fracture zone.

## 1. Introduction

Aluminum alloy has a low density, resistance to low temperature, and good corrosion resistance. Its plasticity, processing properties, mechanical properties, welding properties, forming properties, and surface treatment properties are excellent. Aluminum alloys are widely used in various industries such as aerospace, automobiles, and machinery manufacturing [1]. At present, in addition to gas welding and arc welding, aluminum alloys are generally welded by argon arc welding, resistance welding, diffusion welding, and other methods. Some hard aluminum and super-hard aluminum alloys are welded by new types of methods, including argon arc welding, helium arc welding, and friction stir welding; in addition, some new methods and special welding materials are used [2]. At present, global warming and depleted energy supplies are of importance. In order to reduce vehicle energy consumption and vehicle exhaust emissions, the automotive industry uses aluminum alloy materials to reduce vehicle weight. For the realization of lightweight manufacturing and the improvement of fuel economy, it will be of great significance to solve the problem of welding aluminum and steel, which are dissimilar metals.

Cold metal transfer (CMT), a cold metal over-welding technology, is a new type of welding technology without slag splash that was developed by Fronius International GmbH. It is a high quality technique with reliable and rapid arc ignition. During the welding process, the arc is more stable and arc length control is more accurate [3]. Because of these advantages of CMT technology, it is widely used in the welding of dissimilar metals, such as the welding of dissimilar aluminum alloys, the welding of aluminum alloys and steel, etc. In order to study the mechanisms of damage to aluminum/steel butt joints, this paper uses CMT fusion-brazing technology to connect 6061 aluminum alloy plates and Q235 galvanized steel plates.

In the past, researchers have tried almost all welding methods to weld aluminum and steel. Fukumoto et al. [4] studied friction welding, Hou Fachen et al. [5] investigated explosive welding, and Cakmakkaya et al. [6] explored diffusion welding. These all belong to the category of pressure welding, but with this welding method, it is not easy to control the welding quality and the production efficiency is low. Lv Xueqin et al. [7] studied transition layer brazing, Peng et al. [8] examined vacuum brazing, and Koltsov et al. [9] reported on laser brazing. These all belong to the category of brazing, but the obtained welded joint has low strength and poor resistance to high temperatures. In recent years, fusion-brazing has been studied more frequently; it has the characteristics of both fusion welding and brazing and is suitable for the connection between dissimilar metals with very different melting points. Shi Yu et al. [10] studied MIG fusion brazing, Song et al. [11] researched TIG fusion-brazing, and Dharmendra et al. [12] investigated laser fusion brazing. All these studies reported welded joints with good performance. Fatigue studies on aluminum/steel joints have also been performed. Kowalski M. [13] reported the results of a fatigue crack growth simulation of the transition joint for S235JR steel and A5083 aluminum with a Grade 1 titanium interlayer coat and A1050 aluminum. The crack growth phenomena observed during fatigue testing are also described.

## 2. Experimental

### 2.1. Experimental Materials

The experimental materials were 6061 aluminum alloy plates and Q235 galvanized steel plates; the experimental welding wire was ER4043 (AlSi_5_). Their chemical composition is shown in Table 1, Table 2 and Table 3. Information on their physical properties and mechanical behavior is shown in Table 4. The matrix phase of the 6061 aluminum alloy is α-Al, its key strengthening phase is β (Mg_2_Si), its status is T6 [14], its tensile strength is 332 MPa, and its elongation is 5%. The Q235 steel is composed of α-Fe and pearlite [15], and on its surface there is a 10 μm Zn layer applied by hot-dip coating, with a Zn content of 99.99%. The thickness of both the 6061 aluminum alloy plates and the Q235 galvanized steel plates is 3 mm.

### 2.2. Experimental Methods

#### 2.2.1. Material Pretreatment and Welding

Open a V-shaped groove on the steel side, with an angle of 30°. Mechanically grind the 6061 aluminum alloy plate before welding. Completely remove the surface oxide film and then clean it chemically. Use fine sandpaper to sand the surface of Q235 galvanized steel plate and groove. Wash the area to be welded with acetone before clamping. Control the gap between the two plates to 1.5 mm during clamping. Use the CMT welding machine to perform the welding; control heat input at about 150 J/mm. Set the arcing current to 150% and the arcing current to 50%. Select argon as the protective gas with a flow rate of 15 L/min. The dry elongation is 10 mm and the welding gun inclination angle is 30°. The specified parameters include: welding voltage, 10 V; welding current, 40 A; wire feeding speed, 5.5 m/min; welding speed, 0.5 m/min. The welding diagram is shown in Figure 1.

#### 2.2.2. Observation of Weld Microstructure

After welding, cut the metallographic sample by a wire cutter perpendicular to the weld direction for inlaying, grinding, and polishing. Then, use 1% HF + 1.5% HCl + 2.5% HNO_3_ to corrode the aluminum side and 4% HNO_3_ + 96% C_2_H_5_OH to corrode the steel side. Observe the microstructure by VHK-600K ultra-depth microscope.

#### 2.2.3. Mechanical Properties Test

Prepare the tensile test specimen by a wire electrical discharge machining along the welding direction, as shown in Figure 2. The working principle of the wire electrical discharge machining is shown in Figure 3. Using a moving thin metal wire (copper wire or molybdenum wire) as an electrode, the diameter of the wire is generally between 0.12~0.20 mm. Using thin molybdenum wire as tool electrode for cutting; the wire storage drum makes the molybdenum wire move forward and backward alternately, and the processing energy is supplied by the pulse power supply. Pouring the working fluid medium between the electrode wire and the workpiece, the two coordinate directions of the worktable in the horizontal plane follow the predetermined control program, respectively. According to the state of the spark gap, the servo feed moves to synthesize various curved trajectories to cut the workpiece into shape. Use HXD-1000TCM/LCD Vickers microhardness tester to test the microhardness, the loading force is 100 gf and the holding time is 15 s. Stretch the sample with an AG-25TA electronic universal material testing machine at a stretching speed of 0.1 mm/s.

#### 2.2.4. Fatigue Performance Test and Fatigue Fracture Appearance Observation

In the direction perpendicular to the weld, use the wire cutter to cut the fatigue specimen of the welded joint, as shown in Figure 3, make sure that the weld is in the middle. All fatigue specimens must be sanded and polished before fatigue testing. In the fatigue test, the minimum stress value of the welded joint is set to 40 MPa; the stress level increases by 20 MPa until it breaks. This process takes the form of a sine wave with an initial phase of 270°, a frequency of 1 Hz, a stress ratio of 0, and a cycle number of 105. The experimental data are automatically collected by the software, and complete data from the first five hundred cycles are collected. The subsequent cycles are collected at equal intervals, with one cycle of data collected every fifteen cycles. Finally, observe the fatigue fracture by S-3400N scanning electron microscope.

## 3. Results

### 3.1. Microstructure of Welded Joint

The appearance of the weld joint after welding is shown in Figure 4a. The weld joint is well formed, continuous, and uniform. The pattern on the weld joint resembles fish scales. The shape of the weld joint is high and narrow; the height is 75 mm. A macroscopic image of the welded joint is in Figure 4b. The dotted line indicates the position of the Q235 galvanized steel sheet and the 6061 aluminum alloy base metal (6061BM) before welding. The comparison of the positions before and after welding shows:

Aluminum alloy is fully melted during the welding process, and it solidifies with the melted wire to form the weld zone (WZ), as shown in Figure 5. The eutectic point of Al–Si is 577 °C [17]; during the solidification of the molten pool, α-Al is formed at a temperature higher than 557 °C, and Al–Si is formed at a temperature lower than 557 °C. So the weld zone is composed of α-Al and Al–Si [18]. The arc stirring effect of CMT inhibits the growth of grains to a certain extent, because during the welding process, the arc stirs in the molten pool, so the liquid metal in the molten pool is mixed more uniformly, and the solidified tissue is more uniform. At the same time, the cooling rate of the molten pool is accelerated, the grain nucleation rate is increased, and the grain cannot grow further. In short, under the action of the arc stirring force, the high-temperature melt further from the solidification front and the high-temperature melt near the interface will make a forced exchange with the low-temperature melts with a high percentage of solids, to change the temperature field and concentration field of the melt at the solidification front. Therefore, nucleation and crystallization proceed simultaneously in a wide range within the direction. Solidification crystallization is destroyed and the dynamic crystallization is strengthened. In Figure 5a, the center of the weld seam is mostly small equiaxed crystals that are tightly arranged, and its directionality is not obvious. As shown in Figure 5b, near the heat-affected zone, part of the equiaxed crystals transform into coarse columnar crystals, and its directionality is obvious.

During the welding process, the steel does not melt. As shown in Figure 6a, the steel connects directly with the molten liquid alloy in the weld seam to form the Fe–Al intermetallic compound layer (IMC), which also called interface layer. In Figure 6b, the thickness of the interface layer is 10 μm. The interface layer is tongue-shaped on the steel side and needle-shaped on the aluminum side.

The aluminum alloy and steel close to the welding seam have significant changes in organization and performance, and then become the welding heat-affected zone (HAZ). The heat-affected zone near the weld is heated higher, and the low-melting-point strengthening phase melts. In Figure 7a, the low-melting-point strengthening phase gathers at the grain boundary, and forms coarse equiaxed crystals after cooling and precipitation. At the junction of the weld zone and the heat-affected zone, there is an area with coarse grains and uneven structure; this is the fusion zone. The range of the fusion zone is extremely narrow; it is often called the fusion line. The heat-affected zone near the aluminum alloy is less heated, and most of the columnar low-melting-point precipitation phase melts and agglomerates after cooling. As shown in Figure 7b, compared with the heat-affected zone close to the weld seam, the grain size of the heat-affected zone close to the aluminum alloy is significantly increased and become coarse columnar crystals. We can see the characteristics of competitive growth and epitaxial solidification.

The structure of the heat-affected zone on the steel side is shown in Figure 8. The temperature during welding is between 750 °C and 900 °C; the steel near the weld seam has reached the temperature of incomplete annealing. As seen in Figure 8b, some black martensite appears. The grain refinement is not complete; only part of the pearlite is transformed into austenite, so the small equiaxed crystals and the black aggregates are mixed. As shown in Figure 8a, under the influence of the temperature gradient, the grains become smaller and the black aggregates become more obvious closer to the interface layer.

The weld zone, the interface layer, and the welding heat-affected zone finally form the welded joint.

### 3.2. Microhardness and Stretchability of Welded Joint

The microhardness test results of the welded joint are shown in Figure 9. The low arc heat input of the CMT welding causes few effects on the microstructure of the steel, so the hardness of the steel does not change much; it is about 143 HV. When the interface layer is reached, the microhardness suddenly rises to the maximum (280 HV). The heat-affected zone near the aluminum side has the lowest microhardness (43 HV) due to grain coarsening. The minimum microhardness of the weld zone is 59 HV, which is lower than the microhardness of the aluminum alloy (about 70 HV). This indicates that due to the back dissolution and re-precipitation of the strengthening phase, a softening zone occurs.

Figure 10 shows the tension curve of the welded joint. When the load is 6.15 kN and the displacement is 0.715 mm, the yield point is reached. When the load drops to 5.95 kN, the curve begins to rise until the load reaches the maximum value of 6.65 kN; at this moment, the displacement is 0.910 mm, which is the breaking point. According to the curve, the tensile strength of the welded joint is 110.83 MPa.

### 3.3. Fatigue Performance of Welded Joint

Figure 11 is the S–N curve of the welded joint. When the stress is greater than 170 MPa, the specimen breaks after the first stress alternation. When the stress is less than 170 MPa, the specimen break after stress alternations that continue for a period of time. When the stress is 70 MPa and below, the fatigue life can reach 10^5^.

Figure 12a–c shows the hysteresis loop of the fatigue coupon at the initial stage, stable stage, and late stage under 130 MPa. Then, representative cycles for calculation in these three stages can be selected:

The first cycle is shown in Figure 12d:$$\begin{array}{l} \sigma_{\max}=132.593\;\mathrm{MPa}\\ \sigma_{\min}=-8.257\;\mathrm{MPa}\\ \Delta \sigma =140.85\;\mathrm{MPa}\\ \Delta \varepsilon =0.406\% \end{array}$$

Under 130 MPa of stress, the fatigue life of the coupon is 18,360 cycles, so the hysteresis loop of cycle 9180 can be considered the stable hysteresis loop. This cycle is shown in Figure 12e:$$\begin{array}{l} \sigma_{\max}=129.904\;\mathrm{MPa}\\ \sigma_{\min}=-0.166\;\mathrm{MPa}\\ \Delta \sigma =130.7\;\mathrm{MPa}\\ \Delta \varepsilon =0.335\% \end{array}$$

The last cycle is shown in Figure 12f:$$\begin{array}{l} \sigma_{\max}=129.814\;\mathrm{MPa}\\ \sigma_{\min}=0.104\;\mathrm{MPa}\\ \Delta \sigma =129.71\;\mathrm{MPa}\\ \Delta \varepsilon =0.360\% \end{array}$$

In the comparison of these three cycles, it can be found that during the initial cycles, the coupon is subjected to cyclic loading, so the hysteresis loop is not completely closed. With the passage of time, the coupon becomes stable after a certain number of cycles, and the hysteresis loop is closed; this is the stable hysteresis loop. In the process, the range of the strain first decreases and then increases, while the flow keeps increasing. This shows that the coupon experiences cyclic hardening from the initial stage to the stable stage, and the coupon experiences cyclic softening from the stable stage to the late stage. Throughout the fatigue process, the hysteresis loop slowly moves to the right with the increase in the number of cycles, indicating that a ratchet effect has occurred.

During this fatigue test, the coupon was broken in two areas: one is the interface layer and the other is the weld zone. This indicates that the fatigue strength limit of these two areas is low. It can be found from Figure 13a that the fatigue fracture is mainly caused by incomplete fusion and porosity near the surface of the coupon. The specific fracture process will be analyzed in the following section. In Figure 13b, many secondary cracks and tearing edges appeared around the indentation during the fatigue test. In Figure 13c, when the main cracks merge to form new cracks, fatigued strips are produced due to slip. The fatigued strips here are jagged and irregular; this is a typical brittle fatigue strip. In Figure 13d, there are dimples in the instantaneous fracture area, and there is porosity in the dimple. However, the depth of the dimple is small, indicating that the toughness of the weld is poor. Overall, the fatigue fracture mode of the welded joint is a ductile–brittle mixed fracture.

## 4. Discussion

### 4.1. Initiation of Fatigue Cracks

Previous experiments have shown that in the fatigue coupon of the welded joint, the fatigue crack initiation is mainly located near the surface of the coupon, and the welding defects include: (1) inclusions and second-phase particles; (2) porosity and incomplete fusion.

Many experimental results have proved that inclusions and second-phase particles (namely entrainment) in high-strength aluminum alloys have a very important effect on the initiation of cracks. There are three main reasons: (1) the interface between the second-phase particle and the matrix is separated, causing the cracks to sprout; (2) sprouting from pre-cracked inclusions; (3) sprouting through the interaction of slip and particles under low stress levels. In Figure 14, before the fatigue test of the coupon, the entrainment is connected to the matrix. After the fatigue test is started, the stress alternation works. One side of the entrainment, which intersects the tensile axis, begins to separate from the matrix; over times, the other side of the entrainment also separates from the matrix like this. As time goes by, the separation zone expands and micro-holes in the nearby matrix are formed. Then, the micro-holes connect to each other to form the microcrack, which is perpendicular to the tensile stress axis. After that, the microcrack on one side of the entrainment expands, while the other side of the entrainment continues to form micropores. During the fatigue process, these micro-holes continue to form and then expand into microcracks.

Generally, fatigue caused by porosity and incomplete fusion are usually because of porosity and imcomplete fusion caused by uneven stress distribution, and concentration appears on the surfaces of porosity and incomplete fusion. In Figure 15, after the fatigue test started, under the action of stress concentration, microintrusions occur and form the slip steps in the first cycle on the edge of the porosity and incomplete fusion. As the number of cycles increases, the shearing stress on the same slip band becomes larger in the opposite direction, and reverse slip occurs in the same slip zone. Then, the same mechanism repeats in each cycle and the cycling slip occurs. Subsequently, slipping steps, intrusions, and extrusions, are formed on the edge of porosity and incomplete fusion, which makes the edge of porosity and incomplete fusion more rough and uneven. At this time, microcracks began to sprout along the slip band. In fact, studies have shown that the intrusion is due to the slight displacement of the slip plane during loading and unloading, and the extrusion is due to the reverse slip appearing on the lower side of the slip band [19].

### 4.2. Extension of Fatigue Cracks

Many microcracks that started to sprout will stop growing immediately, and usually only those with very favorable slip can continue to grow. After the microcracks occur, the microcracks in the slip zone continue to increase along the main slip system. Among them, PSB is formed preferentially on the slip plane where the shear stress is the largest [20,21]. Usually, PSB will lead to the nucleation of microcracks, so microcracks generally expand in a “Z” shape at an angle of 45° to the principal stress direction. When the microcrack’s size range increase to 2–3 grains, the stress intensity factor range ∆K at the tip of the microcrack becomes large enough, which causes the slip of other slip systems to be activated and began to slip. Subsequently, the expansion of the crack begins alternately along the two slip systems in a propagation direction that is perpendicular to the principal stress direction. At this time, the microcracks begin to become macrocracks.

It is worth mentioning that if a crack encounters a hole during its expansion, its expansion direction will change. As shown in Figure 16, due to the existence of holes, stress concentration will occur under the action of stress alternation, which causes many secondary cracks to form around the holes. If the hole is large enough and located close to the surface of the coupon, some secondary cracks nearby, for which the expansion direction tends to be the same as that of the original fatigue crack, will be induced to become a new fatigue crack the initiation under the action of stress alternation. The expansion direction is consistent with that of the original fatigue crack initiation.

### 4.3. Instantaneous Fracture of Fatigue Coupon

The instantaneous fracture zone of fatigue coupon appears as toughness fracture, and the formation of dimples in welded joints is mainly due to impurity defects. According to the dislocation theory [22], there are dislocation loops around the second-phase particles or inclusions. Under the action of stress alternation, the dislocation loop around it will move towards the second-phase particles or inclusions. In Figure 17, when the elasticity strain accumulated in the process can overcome the binding force between the second phase particles and the matrix or inclusions and the matrix, micro-holes are formed. As the cycle progresses, these micro-holes continue to grow and quickly expand until they aggregate. At the same time, the cross-sectional area of the matrix between neighboring micro-holes reduces. When all the micro-holes are connected, instantaneous fracture occurs and the dimpled morphology of the fracture is formed.

### 4.4. Improved Fatigue Performance of Welded Joints

Based on the above discussion, it can be found that defects are the most important factor affecting the fatigue performance of aluminum/steel butt joints in CMT fusion brazing technology. Therefore, the most important part of the welding process is to minimize the occurrence of these defects. First, it is critical to prevent the generation of pores; therefore, we need to limit the incorporation of hydrogen into metals and reduce sources of hydrogen. Second, we need to choose suitable welding materials to obtain a weld joint with good composition. Finally, the welding process must be controlled, and standardized operations must be chosen to avoid or suppress the generation of defects.

## 5. Conclusions

The welded joint composed of the weld zone, the interface layer, and the welding heat-affected zone is well formed and has a height of 75 mm. The HAZ near the aluminum side has the lowest microhardness (43 HV) and the interface layer has the maximum microhardness (280 HV). The tensile strength of the welded joint is 110.83 MPa.The fatigue strength limit of the fatigue coupon is 170 MPa. In the fatigue process the ratchet effect occurs; the coupon first experiences cyclic hardening and then cyclic softening. The coupon was broken at interface layer or weld zone where the fatigue strength limit is low.The fatigue crack initiation is mainly caused by: (1) inclusions and second-phase particles; and (2) porosity and incomplete fusion; only microcracks with very favorable slip can expand. When cracks encounter holes during expansion, the expansion direction will change. The fatigue coupon displays a toughness fracture in the instantaneous fracture zone.The most important part of the welding process is to minimize the occurrence of defects: first, by limiting the incorporation of hydrogen into metals and reduce sources of hydrogen; second, by choosing suitable welding materials; and finally, by controlling the welding process and choosing standardized operations.

## Figures and Tables

**Figure 1 materials-15-02367-f001:**
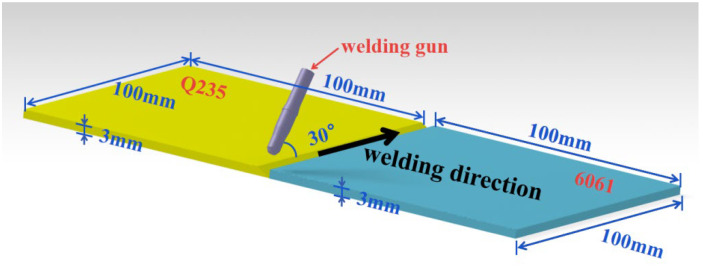
Welding diagram.

**Figure 2 materials-15-02367-f002:**
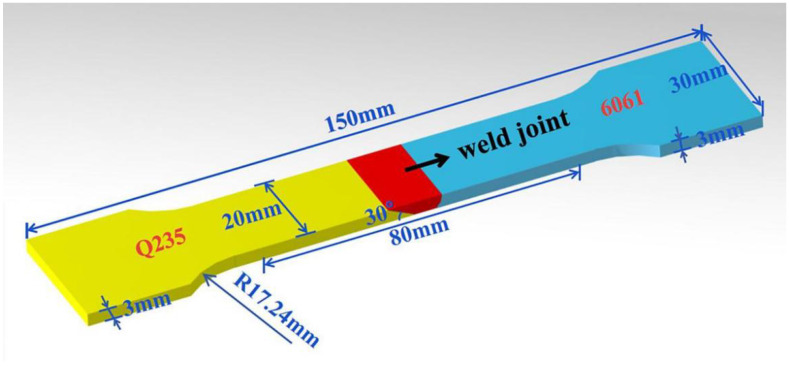
Tension coupon.

**Figure 3 materials-15-02367-f003:**
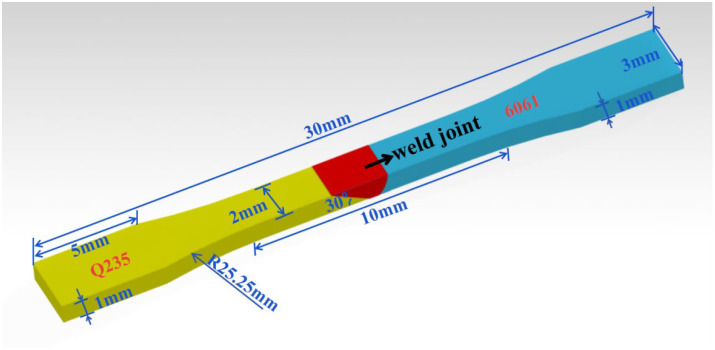
Fatigue coupon.

**Figure 4 materials-15-02367-f004:**
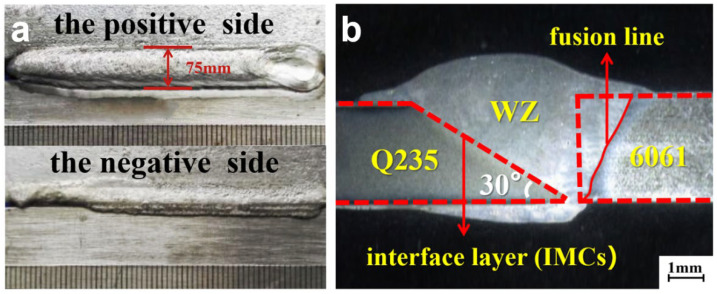
Morphology of welded joint: (**a**) appearance of the weld zone; (**b**) cross-section of the weld zone.

**Figure 5 materials-15-02367-f005:**
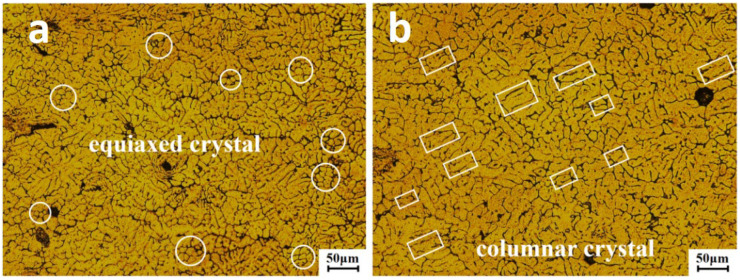
Microstructure of weld zone: (**a**) equiaxed crystals in weld zone; (**b**) columnar crystals in weld zone.

**Figure 6 materials-15-02367-f006:**
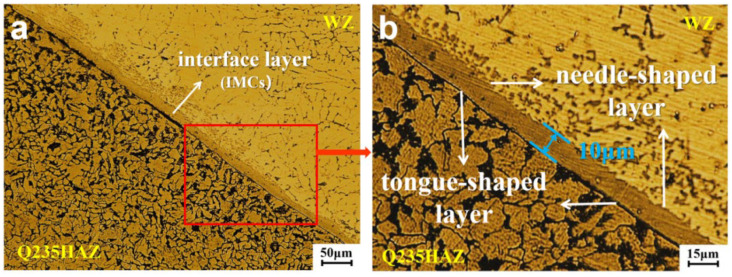
Microstructure of interface layer: (**a**) microstructure under 50 μm; (**b**) microstructure under 15 μm.

**Figure 7 materials-15-02367-f007:**
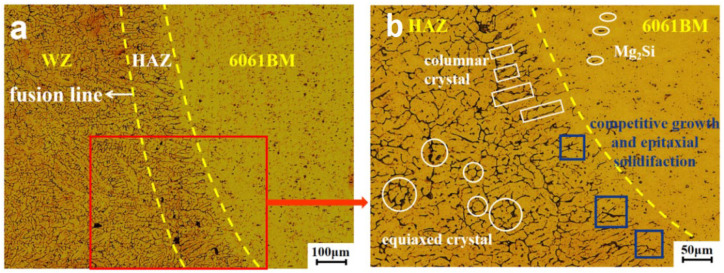
Microstructure of heat-affected zone near the aluminum alloy: (**a**) microstructure under 100 μm; (**b**) microstructure under 50 μm.

**Figure 8 materials-15-02367-f008:**
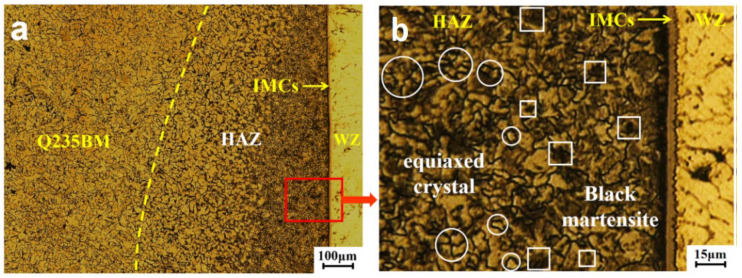
Microstructure of heat-affected zone near the steel: (**a**) microstructure under 100 μm; (**b**) microstructure under 15 μm.

**Figure 9 materials-15-02367-f009:**
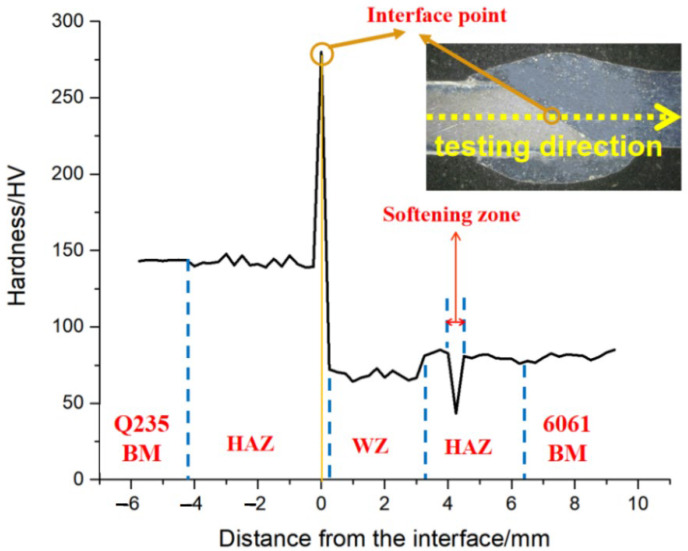
Microhardness of welded joint.

**Figure 10 materials-15-02367-f010:**
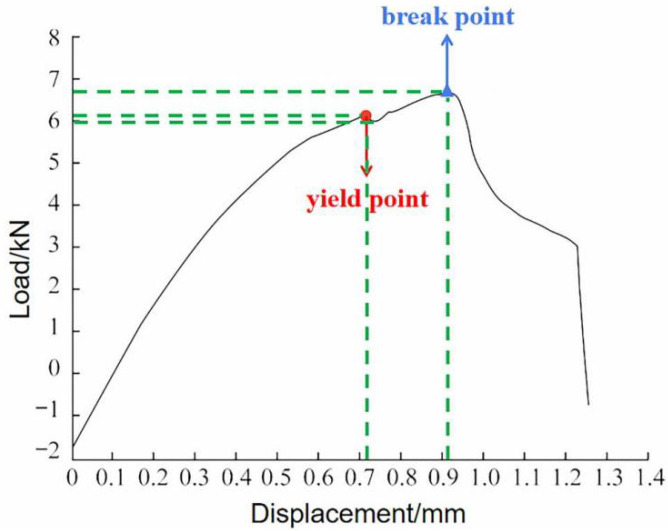
Tension curve of welded joint.

**Figure 11 materials-15-02367-f011:**
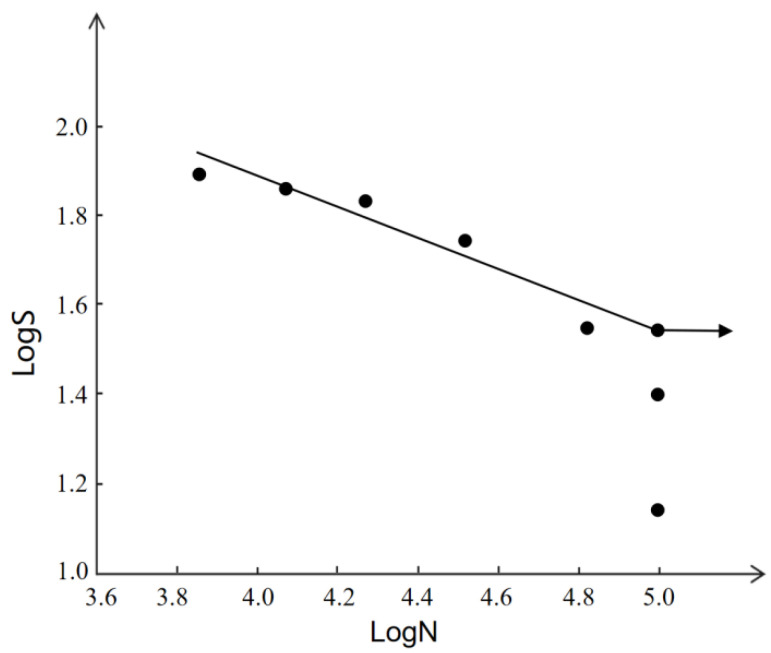
S–N curve of welded joint.

**Figure 12 materials-15-02367-f012:**
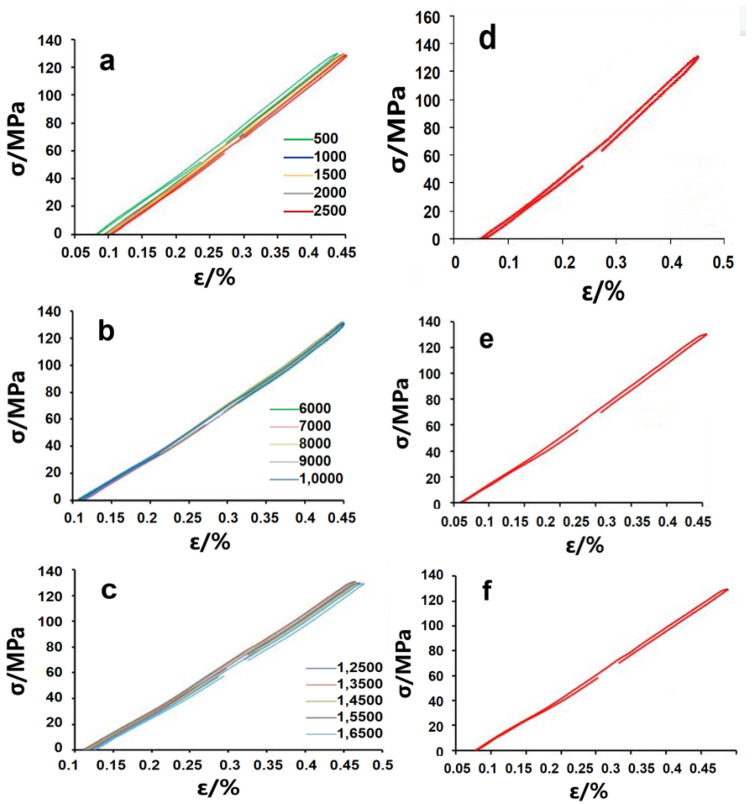
Hysteresis loop under 130 MPa: (**a**) initial stage; (**b**) stable stage; (**c**) late stage; (**d**) first cycle; (**e**) cycle 9180; (**f**) the last cycle.

**Figure 13 materials-15-02367-f013:**
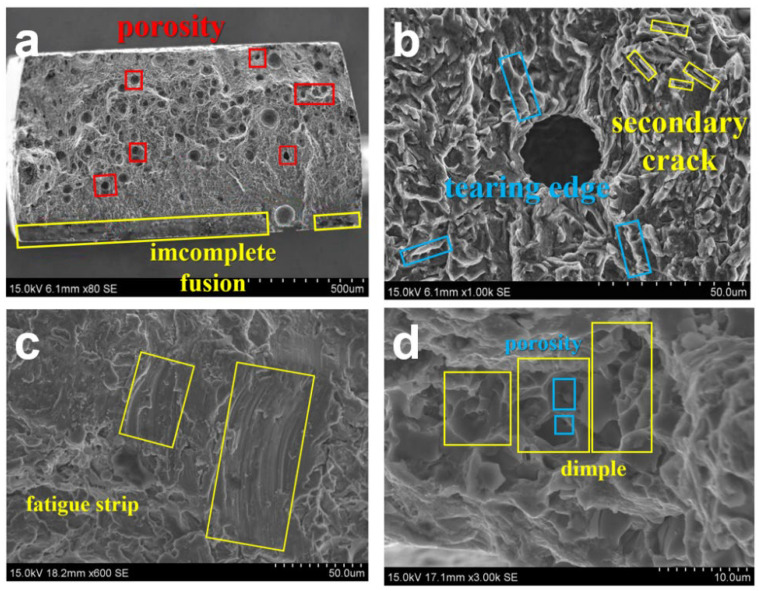
Fatigue fracture micromorphology: (**a**) fatigue fracture source; (**b**) secondary cracks and tearing edges; (**c**) fatigue strip (**d**) dimple and porosity.

**Figure 14 materials-15-02367-f014:**
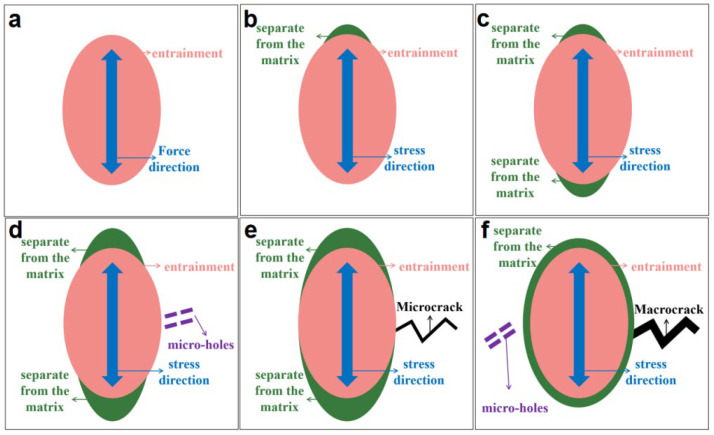
Fatigue crack initiation caused by inclusions and second-phase particles: (**a**) stress alternation effect on entrainment; (**b**) one side separates from the matrix; (**c**) both sides separate from the matrix; (**d**) micro-hole appears; (**e**) microcracks appear; (**f**) micro-holes appear again.

**Figure 15 materials-15-02367-f015:**
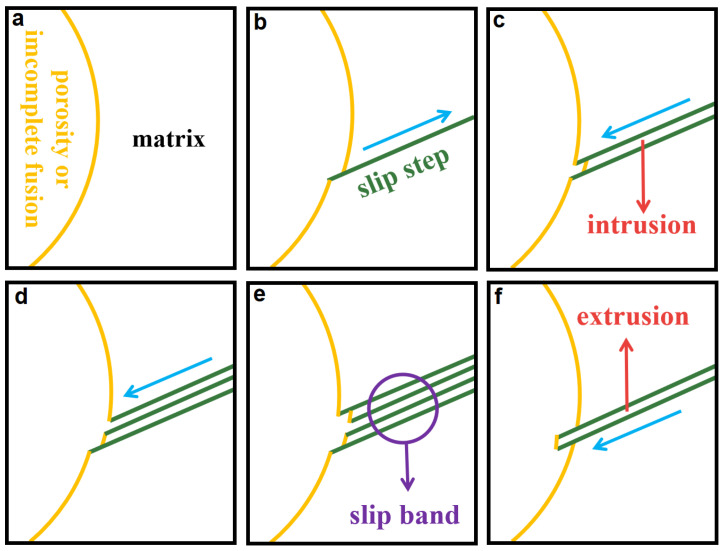
Fatigue crack initiation caused by porosity and incomplete fusion: (**a**) original appearance of porosity and incomplete fusion; (**b**) slip steps; (**c**) intrusion; (**d**) shearing stress on the same slip band becomes larger; (**e**) slip band; (**f**) extrusion.

**Figure 16 materials-15-02367-f016:**
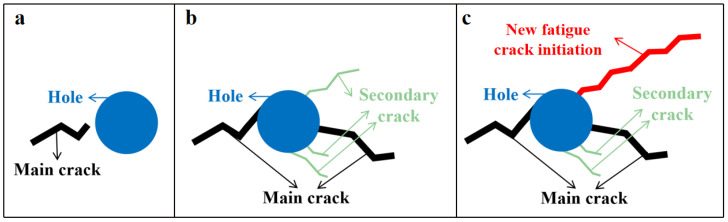
Expansion of cracks when holes are encountered: (**a**) hole and main crack; (**b**) secondary crack; (**c**) new fatigue crack initiation.

**Figure 17 materials-15-02367-f017:**
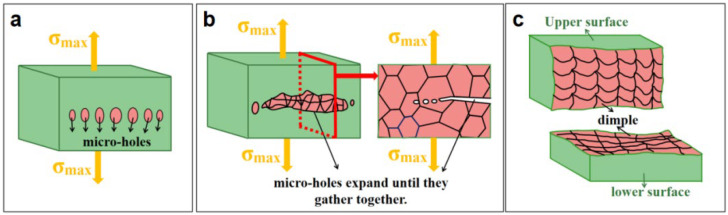
Instantaneous fracture of fatigue coupon: (**a**) micro-holes; (**b**) micro-holes expand and aggregate; (**c**) fracture.

**Table 1 materials-15-02367-t001:** Main chemical composition of 6061 aluminum alloy (%) [16].

Material Content Element	Si	Fe	Cu	Mn	Mg	Zn	Cr	Ti	Al
6061	0.4~0.8	0.7	0.15~0.4	0.15	0.8~1.2	0.25	0.04~0.35	0.15	margin

**Table 2 materials-15-02367-t002:** Main chemical composition of Q235 galvanized steel (%) [16].

Material Content Element	C	Si	Mn	S	P
Q235 (B)	0.12~0.20	≤0.30	0.30~0.70	≤0.45	≤0.45

**Table 3 materials-15-02367-t003:** Main chemical composition of ER4043 welding wire (%) [16].

Material Content Element	Si	Fe	Cu	Mn	Mg	Zn	Ti	Al
ER4043 (AlSi_5_)	4.5~6.0	≤0.8	≤0.30	≤0.05	≤0.05	≤0.10	≤0.2	margin

**Table 4 materials-15-02367-t004:** Physical properties and mechanical behavior of 6061 aluminum alloy, Q235 galvanized steel, and ER4043 welding wire [16].

Material Numerical Value Properties	Density (g·cm^−3^)	Hardness (HV)	Melting Point (°C)	Elongation Rate (%)	Tensile Strength (MPa)	Yield Strength (MPa)
6061	2.73	95	580~650	25.0	290	240
Q235	7.86	140	1500	≥26	370~500	235
ER4043 (AlSi_5_)	-	-	580~620	-	-	-

## Data Availability

Not applicable.

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
