# Peer review of "Study of Microstructure and Fatigue in Aluminum/Steel Butt Joints Made by CMT Fusion-Brazing Technology"

_materials, 2022, doi:10.3390/ma15072367_

Round 1

Reviewer 1 Report

  1. Please explain all abbreviations and markings at work, e.g. D_sigma, D_epsilo, PSB, HAZ, and others.
  2. Page 11, line 8 - is …the stress intensity factor ΔK… should be …the stress intensity factor range ΔK...
  3. Please provide more information about brittle, ductile or mixed cracks?
  4. Fig. 11 - the S-N diagram according to the standard should have at least two results for a given load level.
  5. Fig. 12 - enlarge the descriptions in Fig. 12 because they are hardly legible.
  6. It would also be worthwhile to quote the following paper: 1) Kowalski M., Rozumek D., Numerical simulation of fatigue crack growth in steel-aluminium transition joint. AIP Conference Proceedings, Vol. 2186, No. 170020, 2019.

Author Response

Dear reviewer,

Thank you very much for your valuable comments. All the comments from you have been modified in the text. The following is the specific content of the modification, labels come from newly uploaded files.

  1. Please explain all abbreviations and markings at work, e.g. D_sigma,D_epsilo, PSB, HAZ, and others.

Line10,38,133,136,159,166: cold metal transfer(CMT), weld zone(WZ), 6061 aluminum alloy base metal(6061BM), intermetallic compound (IMC), heat-affected zone(HAZ)

  1. Page 11, line 8 - is …the stress intensity factor ΔK… should be …the stress intensity factor range ΔK...

Line298: When the microcrack’s size range increase to  size of 2-3 grain, the stress intensity factor range ∆K at the tip of the microcrack becomes large enough, which causes the slip of other slip systems to be activated and began to slip.

  1. Please provide more information about brittle, ductile or mixed cracks?

Line238-247: During this fatigue test, the coupon was broken in two areas, one is the interface layer and the other is the weld zone. It indicates that the fatigue-strength limit of these two areas are low. It can be found from Figure13(a) that the fatigue fracture is mainly caused by the imcomplete fusion and porosity near the surface of the coupon. The specific fracture process will be analyzed in the following section.In Figure13(b), many secondary cracks and tearing edge appeared around the indentation during the fatigue test. In Figure13(c), when main cracks merge to form new cracks, fatigue strip are produced due to slip.The fatigue strip here are jagged and irregular, this is a typical brittle fatigue strip. In Figure13(d), there are dimples in the instantaneous fracture area, and there has porosity in the dimple. However, the depth of the dimple is small, indicating that the toughness of the weld is poor. Overall, the fatigue fracture mode of welded joint is ductile-brittle mixed fracture.

  1. 11 - the S-N diagram according to the standard should have at least two results for a given load level.

It is true that we did two groups for each stress level, but during the analysis, we chose the better data for analysis, and did not list all the data.

  1. 12 - enlarge the descriptions in Fig. 12 because they are hardly legible.

Line234

  1. It would also be worthwhile to quote the following paper: 1) Kowalski M., Rozumek D., Numerical simulation of fatigue crack growth in steel-aluminium transition joint. AIP Conference Proceedings, Vol. 2186, No. 170020, 2019.:

Line61-64: Kowalski M.[13] research the results of fatigue crack growth simulation of the transition joint for S235JR steel and A5083 aluminum with the Grade 1 titanium interlayer coat and A1050 aluminum. Describe the crack growth phenomena observed during fatigue testing.

Yours sincerely,

Author of the article

Reviewer 2 Report

The material is interesting but needs to improve the language. English has several problems that need to be improved. I suggest going to a proofreader in linguistics to assess.

  • Regarding to text, spaces before commas must be removed. Appears at various points in the text.
  • Reference can be made using more current references focusing on welding and/or brazing (or mig-brazing) of aluminum with steel.
  • Regarding the experiment, the value of the wire diameter was missing.
  • It would be interesting to explain what the “wire cutter” is and how it is made.
  • In the results, what is “fish scales on the face of the weld.”?
  • It would be interesting to explain how the "...arc stiring effect of CMT..." occurs.
  • As a suggestion, I ask you to consider removing all the arrows in the photos (Figure 5, 7b, 8b, 13). The color related to the detail or the detail that can be placed in the legend is enough for the reader to understand what you want to demonstrate.
  • I don't think the hysteresis plots justify the effect they show because your material is composite. If you are going to keep, put them all on the same scale.
  • Conceptually those topics presented in the discussion are known. It would be interesting to dilute them in the previous results to justify their presentation.

Author Response

Dear reviewer,

Thank you very much for your valuable comments. All the comments from you have been modified in the text. The following is the specific content of the modification, labels come from newly uploaded files. I have made a large number of changes in the article for the content that is not explained in detail. These content are too complicated to list below.

  1. The material is interesting but needs to improve the language. English has several problems that need to be improved. I suggest going to a proofreader in linguistics to assess.

  1. Regarding to text, spaces before commas must be removed. Appears at various points in the text.

  1. Reference can be made using more current references focusing on welding and/or brazing (or mig-brazing) of aluminum with steel.

Line47-64:In the past, researchers have tried almost all welding methods to weld aluminum and steel. Fukumoto et al.[4] research on friction welding, Hou Fachen et al.[5] research on explosive welding and Cakmakkaya et al [6] research on diffusion welding. These all belong to the category of pressure welding, but this welding method is not easy to control the welding quality, and the production efficiency is low. Lu Xueqin et al.[7] research on transition layer brazing, Peng et al.[8] research on vacuum brazing and Koltsov et al.[9] research on Laser Brazing. Thease are all belong to the category of brazing, but the obtained welded joint has low strength and poor high temperature resistance. In recent years, fusion-brazing has been studied more frequently, it has the characteristics of both fusion welding and brazing and is suitable for the connection between dissimilar metals with very different melting points. Shi xu et al.[10] research on MIG fusion-brazing, Song et al.[11] research on TIG fusion-brazing and DhariBendraa et al.[12] research on laser fusion-brazing. All of these studies have resulted in welded joints with good performance. Nowadays, fatigue studies on aluminum/steel joints have also been carried out. Kowalski M.[13] research the results of fatigue crack growth simulation of the transition joint for S235JR steel and A5083 aluminum with the Grade 1 titanium interlayer coat and A1050 aluminum. Describe the crack growth phenomena observed during fatigue testing.

  1. Regarding the experiment, the value of the wire diameter was missing.

Line103: Diameter of the wire is generally between 0.12~0.20 mm.

  1. It would be interesting to explain what the “wire cutter” is and how it is made.

Line101-109: The working principle of the wire electrical discharge machining is shown in Figure3:Using a moving thin metal wire(opper wire or molybdenum wire)as an electrode, the diameter of the wire is generally between 0.12~0.20 mm. Use thin molybdenum wire as tool electrode for cutting, the wire storage drum makes the molybdenum wire move forward and reverse alternately, and the processing energy is supplied by pulse power supply. Pouring the working fluid medium between the electrode wire and the workpiece, the two coordinate directions of the worktable in the horizontal plane follow the predetermined control program respectively. According to the state of the spark gap, the servo feed moves to synthesize various curved trajectories to cut the workpiece into shape.

  1. In the results, what is “fish scales on the face of the weld.”?

Line130: The pattern on the weld joint looks like fish scales.

  1. It would be interesting to explain how the "...arc stiring effect of CMT..." occurs.

Line139-148: The arc stirring effect of CMT inhibits the growth of grains to a certain extent, bacause during the welding process, the arc stirs in the molten pool, so the liquid metal in the molten pool is mixed more uniformly, and the solidified tissue is more uniform. At the same time, the cooling rate of the molten pool is accelerated, the grain nucleation rate is increased, and the grain cannot grow further. In short, under the action of the arc stirring force, the high temperature melt farther from the solidification front and the high temperature melt near the interface will make a forced exchange with the low temperature melts with high percent solids, to change the temperature field and concentration field of the melt at the solidification front. Therefore, nucleation and crystallization proceed simultaneously in a wide range within the directional solidification crystallization is destroyand and the dynamic crystallization is strengthe.

  1. As a suggestion, I ask you to consider removing all the arrows in the photos (Figure 5, 7b, 8b, 13). The color related to the detail or the detail that can be placed in the legend is enough for the reader to understand what you want to demonstrate.

Line154, 177, 187, 248

  1. I don't think the hysteresis plots justify the effect they show because your material is composite. If you are going to keep, put them all on the same scale.

Line234

  1. Conceptually those topics presented in the discussion are known. It would be interesting to dilute them in the previous results to justify their presentation.

Yours sincerely,

Author of the article

Reviewer 3 Report

  1. Decipher the abbreviation CMT in the text
  2. Indicate from which literary sources the data in Tables 1-3 were taken.
  3. Check the sentence on line 103.
  4. In the captions to Figures 4-8, give a description of Figures a and b. Also give a description of the sub figures for the figures 12, 13, 14, 15, 16, 17
  5. What is the difference between Figures 14a and 14b?
  6. It is recommended to discuss how the obtained results can be applied in engineering practice.

Author Response

Dear reviewer,

Thank you very much for your valuable comments. All the comments from you have been modified in the text. The following is the specific content of the modification, labels come from newly uploaded files.

  1. Decipher the abbreviation CMT in the text

 Line10, 38: CMT:cold metal transfer

  1. Indicate from which literary sources the data in Tables 1-3 were taken.

Line76, 78, 79: Fatigue damage study of cold metal transition fusion-brazed aluminium/steel dissimilar joints[J]. Science and Technology of Welding and Joining, 2020, 25(4):265-272.

  1. Check the sentence on line 103.

Line129-131: The appearance of the weld joint after welding is shown in Figure4(a). It can be seen that the weld joint is well formed and is continuous and uniform. The pattern on the weld joint looks like fish scales.

  1. In the captions to Figures 4-8, give a description of Figures a and b. Also give a description of the sub figures for the figures 12, 13, 14, 15, 16, 17

Line153, 155, 163, 178, 188, 235, 249, 271, 288, 311, 325

  1. What is the difference between Figures 14a and 14b?

Line270:

  1. It is recommended to discuss how the obtained results can be applied in engineering practice.

Line327-335

4.4 Improve fatigue performance of welded joints

Based on the above discussion, it can be found that defects are the most important factor affecting the fatigue performance of aluminum/steel butt joints in CMT fusion-brazing technology. Therefore, the most important thing in the welding process is to minimize the occurrence of these defects. First, the most important thing is to prevent the generation of pores, we need to limit the incorporation of hydrogen into metals and reduce sources of hydrogen; Secondly, we need to choose suitable welding materials to get a weld joint with good composition; Finally, the welding process must be controlled, and and choose standardized operations to avoid or suppress the generation of defects.

Yours sincerely,

Author of the article
